# Inpatient Outcomes of Patients Undergoing Robot-Assisted versus Laparoscopic Radical Cystectomy for Bladder Cancer: A National Inpatient Sample Database Study

**DOI:** 10.3390/jcm13030772

**Published:** 2024-01-29

**Authors:** Le-Wei Fan, Yun-Ren Li, Cheng-Mu Wu, Kai-Ti Chuang, Wei-Chang Li, Chung-Yi Liu, Ying-Hsu Chang

**Affiliations:** 1Division of Urology, Department of Surgery, New Taipei Municipal TuCheng Hospital, Chang Gung Memorial Hospital and Chang Gung University, New Taipei 236, Taiwana9313@cgmh.org.tw (W.-C.L.); 8902087@cgmh.org.tw (C.-Y.L.); 2Department of Biotechnology and Laboratory Science in Medicine, National Tang Ming Chiao Tung University, Taipei 112, Taiwan; 3Division of Plastic and Reconstructive Surgery, Department of Surgery, New Taipei Municipal TuCheng Hospital, Chang Gung Memorial Hospital and Chang Gung University, New Taipei 236, Taiwan

**Keywords:** bladder cancer, in-hospital outcome, nationwide inpatient sample (NIS), radical cystectomy, robot-assisted surgery

## Abstract

**Background:** Bladder cancer is a common urinary tract malignancy. Minimally invasive radical cystectomy has shown oncological outcomes comparable to the conventional open surgery and with advantages over the open procedure. However, outcomes of the two main minimally invasive procedures, robot-assisted and pure laparoscopic, have yet to be compared. This study aimed to compare in-hospital outcomes between these two techniques performed for patients with bladder cancer. **Methods:** This population-based, retrospective study included hospitalized patients aged ≥ 50 years with a primary diagnosis of bladder cancer who underwent robot-assisted or pure laparoscopic radical cystectomy. All patient data were extracted from the US National Inpatient Sample (NIS) database 2008–2018 and were analyzed retrospectively. Primary outcomes were in-hospital mortality, prolonged length of stay (LOS), and postoperative complications. **Results:** The data of 3284 inpatients (representing 16,288 US inpatients) were analyzed. After adjusting for confounders, multivariable analysis revealed that patients who underwent robot-assisted radical cystectomy had a significantly lower risk of in-hospital mortality (adjusted OR [aOR], 0.50, 95% CI: 0.28–0.90) and prolonged LOS (aOR, 0.63, 95% CI: 0.49–0.80) than those undergoing pure laparoscopic cystectomy. Patients who underwent robot-assisted radical cystectomy had a lower risk of postoperative complications (aOR, 0.69, 95% CI: 0.54–0.88), including bleeding (aOR, 0.73, 95% CI: 0.54–0.99), pneumonia (aOR, 0.49, 95% CI: 0.28–0.86), infection (aOR, 0.55, 95% CI: 0.36–0.85), wound complications (aOR, 0.33, 95% CI: 0.20–0.54), and sepsis (aOR, 0.49, 95% CI: 0.34–0.69) compared to those receiving pure laparoscopic radical cystectomy. **Conclusions:** Patients with bladder cancer, robot-assisted radical cystectomy is associated with a reduced risk of unfavorable short-term outcomes, including in-hospital mortality, prolonged LOS, and postoperative complications compared to pure laparoscopic radical cystectomy.

## 1. Introduction

Bladder cancer is the most common malignancy of the urinary tract [1]. The Global Cancer Statistics Report stated that 573,278 individuals were diagnosed with bladder cancer worldwide in 2020, representing 3% of all malignancies worldwide, and 6% in the US [2]. The majority of patients (about 90%) are diagnosed at age 55 years or older [3]. Open radical cystectomy has long been the main surgical treatment for non-metastatic bladder cancer, which is associated with perioperative complication rates of 15~50% and a 3-month mortality rate of 3% [4,5].

Advances in surgical equipment and techniques have led to an increased use of minimally invasive radical cystectomy in the surgical treatment of bladder cancer. Comparable oncological outcomes have been reported between minimally invasive and open radical cystectomy [6,7,8,9,10,11]. When compared to open radical cystectomy, robot-assisted radical cystectomy [6,7,8] and pure laparoscopic radical cystectomy [9,10,11] both had much lower hospital stays, less blood loss, lower transfusion rates, and fewer major perioperative complications. However, whether robot-assisted or pure laparoscopic radical cystectomy is associated with better short- and long-term outcomes remains to be elucidated. The objective of this study was to compare the inpatient outcomes of patients with bladder cancer who underwent either robot-assisted or laparoscopic radical cystectomy, utilizing a nationally representative US inpatient database.

## 2. Materials and Methods

### 2.1. Data Source

This population-based, retrospective observational study extracted all data from the US National Inpatient Sample (NIS) database, which is the largest continuous inpatient care database in the United States, including about 8 million hospital stays each year [12]. The database is administered by the Healthcare Cost and Utilization Project (HCUP) of the US National Institutes of Health (NIH). The patient data consist of primary and secondary diagnoses, primary and secondary procedures, admission and discharge status, patient demographics, projected payment source, hospital stay duration, and hospital characteristics (i.e., bed size/location/teaching status/hospital area). Initial consideration is given to all hospitalized patients for inclusion. The continuously updated, annual NIS database contains patient information from around 1050 hospitals in 44 states, representing a stratified sample of 20% of US community hospitals as defined by the American Hospital Association.

### 2.2. Ethics Statement

This study complies with the terms of the NIS data-use agreement. Given that this study solely involved the analysis of secondary data, there was no direct involvement of the general public or patients. It was granted exemption from requiring IRB approval.

### 2.3. Study Population

Hospitalized patients aged ≥ 50 years with a primary diagnosis of bladder cancer who underwent either pure laparoscopic radical cystectomy or robot-assisted radical cystectomy between 2008 and 2018 were included. Patients with metastatic disease, missing outcomes of interest, and/or missing weight values of the NIS dataset were excluded. Patients were identified in the NIS database using diagnostic and procedure codes of the International Classification of Diseases, Ninth Revision and Tenth Revision, Clinical Modification (ICD-9-CM, ICD-10-CM) as follows: Bladder cancer (ICD-9-CM code: 188.0–188.6, 188.8, 188.9; ICD-10-CM code: C67.0–C67.6, C67.8, F67.9), pure laparoscopic procedure (ICD-9-CM: 57.4–57.7 combined with procedure codes 54.19, 54.21, or 54.51; ICD-10 procedure codes: 0TTB4ZZ, 0TTC4ZZ) or robot-assisted procedure (ICD-9-CM: 57.4–57.7 combined with procedure codes 17.42–17.44, or 17.49; ICD-10 procedure codes: 0TTB4ZZ, 0TTC4ZZ combined with 8E0W3CZ, 8E0W4CZ, 8E0W7CZ, or 8E0WXCZ); metastatic disease (ICD-9-CM: 196.0, 199.1, CM_METS = 1; ICD-10-CM: C77.0-C80.2).

### 2.4. Outcomes

Primary study outcomes were in-hospital mortality, prolonged length of stay (LOS), and postoperative complications. In-hospital mortality data were identified from patients’ hospital discharge disposition records. Hospital LOS was calculated by subtracting the admission date from the discharge date. Postoperative complications, including bleeding, pneumonia, infection, sepsis, and wound complications, were identified using ICD codes, as previously described [6,7,8,9,10,11].

### 2.5. Covariates

Patients’ demographic data, including age, sex, race, and family income-to-poverty ratio, were extracted from the NIS database. Hospital-related characteristics (bed size and location/teaching status) were extracted from the database as part of the comprehensive data available for all participants in accordance with other NIS studies in the medical literature.

### 2.6. Statistical Analysis

Since the NIS database covers a 20% sample of the USA annual inpatient admissions, weighted samples (before 2011 using TRENDWT and after 2012 using DISCWT), stratum (NIS_STRATUM), cluster (HOSPID) were used to produce national estimates for all analyses. The SURVEY procedure in SAS performs analysis for sample survey data. Descriptive statistics of bladder cancer patients undergoing either robot-assisted or laparoscopic radical cystectomy are presented as numbers (*n*) and weighted percentages (%) or mean and standard error (mean ± SE). Categorical data were analyzed by PROC SURVEYFREQ statement and continuous data were analyzed by PROC SURVEYREG statement. Odds ratios (ORs) and 95% confidence intervals (CIs) were calculated for outcomes, including in-hospital mortality, prolonged LOS and major postoperative complications, using logistic regression with the PROC SURVEYLOGISTIC procedure. Covariates with significant differences between the two groups in univariable regression analysis were considered possible confounders and were adjusted in multivariable regression analysis. All *p* values were two-sided and *p* < 0.05 was considered statistically significant. All statistical analyses were performed using the statistical software package SAS software version 9.4 (SAS Institute Inc., Cary, NC, USA).

## 3. Results

### 3.1. Study Population Selection

A total of 4084 patients with diagnosis of bladder cancer who underwent robot-assisted or pure laparoscopic radical cystectomy between 2008 and 2018 were identified in the NIS database. Patients with metastatic disease or having missing information on outcomes, study variables and weight values (n = 800) were excluded. Finally, 3314 patients were included as the analytic sample (representing 16,437 US hospital inpatients) (Figure 1).

### 3.2. Characteristics of the Study Population

Patients’ demographic and clinical characteristics are summarized in Table 1. Patients’ mean age was 69.5 ± 0.2 years, and the majority were males (81.6%) and Whites (86%). A majority of patients had high household income (26.6%), Medicare/Medicaid insurance status (67.6%), and Charles comorbidity index (CCI) scores of 2–3 (71.0%). Most patients were admitted to larger hospitals (66.1%) and urban teaching status (87.6%). Compared to patients who underwent pure laparoscopic radical cystectomy, patients who underwent robot-assisted radical cystectomy were younger (mean age 69.4 vs. 70.6 years, *p* = 0.023), with a higher percentage of males (82.1% vs. 76.8%, *p* = 0.015), higher household income (27.3% vs. 20.3%, *p* = 0.018) and lower percentages of Medicare/Medicaid insurance status (66.8% vs. 74.6%, *p* = 0.021), and more CCI scores of 2–3 (71.5% vs. 67.0%, *p* = 0.017).

Among outcomes, the robot-assisted group had significantly lower in-hospital mortality (1.4% vs. 3.3%, *p* = 0.004) and prolonged LOS (25.5% vs. 32.7%, *p* = 0.004) than the pure laparoscopic group. Percentages of major complications were also significantly lower in the robot-assisted group than in the pure laparoscopic group, including pneumonia (1.8% vs. 3.6%, *p* = 0.009), wound complications (3.3% vs. 6.8%, *p* < 0.001), and sepsis (6.0% vs. 11.7%, *p* < 0.001).

### 3.3. Risk of in-Hospital Mortality and Prolonged LOS between Robot-Assisted versus Pure Laparoscopic Cystectomy

Table 2 and Appendix A show associations between the in-hospital outcomes (mortality and prolonged LOS) and study variables, including the types of minimally invasive surgical procedures. Multivariable analysis revealed that patients who underwent robot-assisted radical cystectomy had significantly lower odds of in-hospital mortality (adjusted OR [aOR], 0.50, 95% CI: 0.28–0.90) and prolonged LOS (aOR, 0.63, 95% CI: 0.49–0.80) than patients who underwent the pure laparoscopic procedure.

### 3.4. Postoperative Complications of Robot-Assisted vs. Pure Laparoscopic Cystectomy

Table 3 and Appendix A show associations between postoperative complications and the two procedures. After adjusting for confounders identified in univariable analysis, multivariable analysis revealed that the robot-assisted procedure was significantly associated with lower risk of complications (aOR, 0.69, 95% CI: 0.54–0.88), bleeding (aOR, 0.73, 95% CI: 0.54–0.99), pneumonia (aOR, 0.49, 95% CI: 0.28–0.86), infection (aOR, 0.55, 95% CI: 0.36–0.85), wound complications (aOR, 0.33, 95% CI: 0.20–0.54), and sepsis (aOR, 0.49, 95% CI: 0.34–0.69) than the pure laparoscopic procedure.

## 4. Discussion

Results of the present study showed that among patients with bladder cancer who underwent minimally invasive radical cystectomy, those who received the robot-assisted procedure had an approximately 50% lower risk of in-hospital mortality and about a 40% lower risk of prolonged LOS compared to those who underwent pure laparoscopic cystectomy. In addition, robot-assisted radical cystectomy was associated with a significantly lower risk for postoperative complications such as bleeding, pneumonia, infection, wound complications, and sepsis compared to pure laparoscopic radical cystectomy. These findings highlight the advantages of robotic-assisted minimally invasive procedures over pure laparoscopic procedures performed for bladder cancer.

Results of the present study are fairly consistent with findings of previous studies for the primary inpatient outcomes of our study—inpatient mortality, hospital LOS and postoperative complications, which were all lower in patients who received robot-assisted cystectomy than in those receiving pure laparoscopic cystectomy. Generally, inpatients in the present study who underwent the robot-assisted procedure were younger, with more males than females, and lower CCI scores (2–3, indicating fewer comorbidities) than in patients receiving the pure laparoscopic procedure. The between-group differences in these patient characteristics, although not significant, may still help to explain certain advantages of robot-assisted procedures over pure laparoscopic procedures as noted in the present and previous studies.

Morbidity and mortality rates for radical cystectomy performed for bladder cancer have been notably high for the complex open procedure, namely 15~50% for perioperative complications and a 3-month mortality rate of 3% [4,5]. However, with quality improvements in surgery over time, including surgical equipment, anesthesia, advanced imaging and increased surgeon experience with the newly introduced minimally invasive cystectomy procedures—including robot-assisted and pure laparoscopic—short-term mortality rates have been falling [3,4]. Most patients (about 90% in US) with bladder cancer are older adults with a mean age of 73 years at diagnosis [13]. Recent evidence strongly supports the causal potential of associations between smoking and bladder cancer [14,15], making it the most significant risk factor for bladder cancer. Older adults with a smoking habit are subject to having more comorbidities involving smoking-related effects, which along with older age and a longer period of exposure of urothelial cells lining of the bladder to mutagenic toxins [16], may also increase their risk for perioperative complications such as pneumonia [17]. Smoking is increasing worldwide, driving the increasing incidence of bladder cancer even while mortality rates decrease [3]. SEER data (https://seer.cancer.gov/explorer/ (accessed on 2 January 2023)) show that the five-year survival rate for bladder cancer is 77% in the US (increased over four decades from 71.9% in 1975), including 95% for in situ disease (51% of cases), 69.5% for localized disease, 7% for regional disease and 4.6% for metastatic disease, demonstrating the poorest prognosis for regional and metastatic disease in bladder cancer patients.

In the present study, hospital LOS was significantly shorter in patients who received the robot-assisted procedure than in those receiving the pure laparoscopic procedure, most likely due to the faster recovery and fewer complications as reported in previous study of the application of robotics in various types of surgery [18]. Surgeons benefit from the advanced computer interface used in robotic-assisted surgery, which improves their depth perception, dexterity and movement control within the operative field. These benefits to surgeons are passed on to patients in the form of faster recovery and fewer complications, which translate into shorter hospital LOS. However, the duration of the robot-assisted procedure, which was not compared between groups in the present study, is reported to exceed that of conventional laparoscopic surgery, without unduly influencing perioperative complications associated with the robot-assisted procedure over time [19].

Postoperative complications evaluated in the present study included bleeding, pneumonia, infection, sepsis, and wound complications. Results showed that, of these, bleeding, pneumonia, infection, sepsis, and wound complications occurred significantly less often in the robotic surgery group compared to the extent of these complications in the pure laparoscopy group. This is consistent with prior results showing that the robotic approach had reduced intraoperative blood loss and transfusion rates compared to open and other laparoscopic approaches [4]. Comparison between outcomes of the robotic approach and those of the open approach showed a 50% reduction in bleeding in the robot-assisted surgeries intra- and postoperatively and significantly lower transfusion rates perioperatively; risk of bleeding was reduced by 19% [8]. Regarding sepsis, while we found significant differences between the two laparoscopic procedures, previous studies compared the robotic or conventional laparoscopic procedures only with results of the open radical cystectomy, reporting more favorable results for ileus and septic complications in the laparoscopic procedure [9]. Otherwise, significant differences were not found in that study in major complications, even though the risk of postoperative morbidity has been reported to be as high as 40–65% for the open procedure [20]. Nevertheless, some studies have reported lower complication rates for both conventional laparoscopy and the robotic procedure. Tang et al. [11], for example, reported a significantly lower risk of complications in patients receiving the laparoscopic procedure. Clearly, more evidence is needed to help determine differences in short- and long-term outcomes between the laparoscopic procedures used currently. Patient selection may be a factor as well as surgeon experience and expertise with radical cystectomy—a complex procedure regardless of the approach used.

### Strengths and Limitations

The main strength of the present study was the use of comprehensive patient data from a large nationally representative inpatient database as the analytic sample, which gives researchers an opportunity to evaluate a broad range of divergent cases from multiple centers, adding credence to the study results. Nevertheless, this study has several limitations. Firstly, it is inherently limited by its retrospective cross-sectional design conducted in the US population, which may limit the generalization of results to other populations and does not allow inferences of causality. Accordingly, election bias also cannot be ruled out. Coding errors are possible as in other studies that used ICD code systems. Secondly, the NIS database did not include outcome-related variables such as preoperative performance status, American Society of Anesthesiologists (ASA) scores, systemic therapy received, and information about postoperative care, such as Enhanced Recovery After Surgery (ERAS) protocols, thus these factors could not be analyzed. Because data on tumor characteristics that may affect outcomes were lacking, including lymph node invasion and the extent of lymph node dissection [21], these clinical factors could not be included in analysis. The type of urinary diversion employed and the details regarding the techniques used for diversion configuration—whether they were totally intracorporeal or extracorporeal—is crucial [22]; however, our study lacks such information due to not available in the dataset. Additionally, the enrollment period spanned a wide range, where more earlier cases were conducted through a purely laparoscopic method and more recent cases employed a robotic approach. Although we have adjusted for admission year in our analyses, it is advisable for readers to interpret the study’s results cautiously. The historical context of these two procedures and changes in the quality of care over time could potentially influence the analyses and create confounding factors. Lastly, since the NIS reports inpatient data only up to discharge, long-term outcomes such as complications, readmission, and survival could not be evaluated. Collecting post-discharge data, with a proper follow-up in the future could provide valuable insights into the long-term outcomes.

## 5. Conclusions

In patients with bladder cancer undergoing minimally invasive cystectomy, robot-assisted radical cystectomy is associated with a reduced risk for in-hospital mortality, prolonged LOS, and postoperative complications, including pneumonia, sepsis, and wound complications, compared to pure laparoscopic radical cystectomy. Further multicenter, prospective study of patients with bladder cancer is still needed to confirm these findings and demonstrate reproducibility, particularly for rates of perioperative complications between surgical approaches.

## Figures and Tables

**Figure 1 jcm-13-00772-f001:**
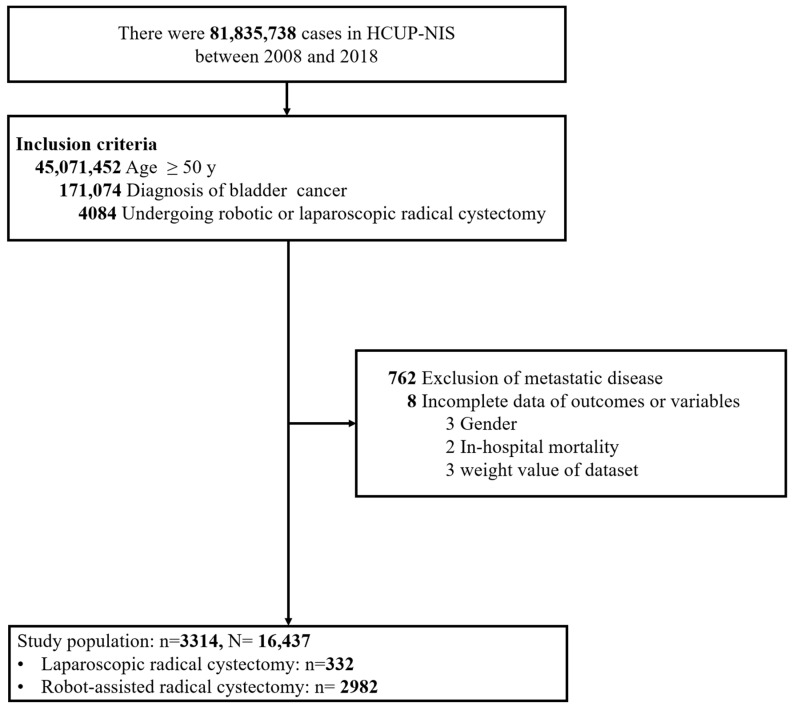
Flow chart of patient selection.

**Table 1 jcm-13-00772-t001:** Demographic and clinical characteristics of the study population.

Characteristic	Total	Laparoscopic Radical Cystectomy	Robot-Assisted Radical Cystectomy	*p*-Value
	(*n* = 3314)	(*n* = 332)	(*n* = 2982)	
In-hospital mortality	54 (1.6)	11 (3.3)	43 (1.4)	0.004
Prolonged LOS ^a,b^	855 (26.2)	105 (32.7)	750 (25.5)	0.004
Complication (any)	834 (25.0)	84 (25.1)	750 (25.0)	0.955
Bleeding	598 (17.9)	53 (15.9)	545 (18.1)	0.279
Pneumonia	65 (1.9)	12 (3.6)	53 (1.8)	0.009
Infection	240 (7.2)	29 (8.5)	211 (7.1)	0.315
Wound complications	122 (3.7)	23 (6.8)	99 (3.3)	<0.001
Sepsis	217 (6.6)	39 (11.7)	178 (6.0)	<0.001
Age	69.5 ± 0.2	70.6 ± 0.4	69.4 ± 0.2	0.023
50–59	485 (14.6)	43 (12.9)	442 (14.8)	0.018
60–69	1146 (34.6)	96 (28.9)	1050 (35.2)	
70–79	1201 (36.3)	142 (42.8)	1059 (35.6)	
80+	482 (14.5)	51 (15.3)	431 (14.4)	
Gender				0.015
Male	2704 (81.6)	255 (76.8)	2449 (82.1)	
Female	610 (18.4)	77 (23.2)	533 (17.9)	
Race				0.078
White	2626 (85.9)	267 (86.4)	2359 (85.9)	
Black	166 (5.5)	23 (7.6)	143 (5.2)	
Hispanic	106 (3.5)	10 (3.2)	96 (3.5)	
Others	156 (5.2)	9 (2.9)	147 (5.4)	
Missing	260	23	237	
Household income				0.018
Quartile 1	606 (18.5)	62 (18.6)	544 (18.5)	
Quartile 2	870 (26.6)	104 (31.8)	766 (26.0)	
Quartile 3	923 (28.3)	96 (29.3)	827 (28.2)	
Quartile 4	869 (26.6)	67 (20.3)	802 (27.3)	
Missing	46	3	43	
Insurance status				0.021
Medicare/Medicaid	2238 (67.6)	246 (74.1)	1992 (66.8)	
Private including HMO	975 (29.4)	77 (23.2)	898 (30.1)	
Self-pay/no-charge/other	101 (3.0)	9 (2.7)	92 (3.1)	
CCI				0.017
0–1	0 (0.0)	0 (0.0)	0 (0.0)	
2–3	2355 (71.0)	223 (67.0)	2132 (71.5)	
4–5	712 (21.5)	73 (22.1)	639 (21.4)	
6+	247 (7.5)	36 (10.8)	211 (7.1)	
Hospital bed size				0.012
Large	2192 (66.1)	210 (63.2)	1982 (66.4)	
Medium	673 (20.5)	88 (26.6)	585 (19.8)	
Small	447 (13.4)	34 (10.2)	413 (13.8)	
Missing	2	0	2	
Hospital location/teaching status				<0.001
Urban teaching	2901 (87.6)	274 (82.6)	2627 (88.2)	
Urban nonteaching	365 (11.0)	49 (14.9)	316 (10.6)	
Rural	46 (1.3)	9 (2.6)	37 (1.2)	
Missing	2	0	2	
Year admission				<0.001
2005–2010	410 (12.2)	42 (12.5)	368 (12.1)	
2011–2014	1165 (34.9)	42 (12.3)	1123 (37.5)	
2015–2018	1739 (52.9)	248 (75.2)	1491 (50.4)	

Continuous variables are presented as mean ± SE; categorical variables are presented as unweighted counts (weighted percentage). LOS, length of stay; HMO, Health Maintenance Organization; CCI; Charlson Comorbidity Index. ^a^ Patients excluded for in-hospital mortality. ^b^ LOS > 9 days. *p*-values < 0.05 are shown in bold.

**Table 2 jcm-13-00772-t002:** Associations between study variables and short-term outcomes in-hospital mortality and prolonged LOS.

Outcomes	Robot-Assisted vs. Pure Laparoscopic			
	Univariate		Multivariate	
	OR (95% CI)	*p* Value	aOR (95% CI)	*p* Value
In-hospital mortality	0.43 (0.23, 0.78)	**0.006**	0.50 (0.28, 0.90)	**0.020**
Prolonged LOS ^a,b^	0.68 (0.53, 0.86)	**0.001**	0.63 (0.49, 0.80)	**<0.001**

^a^ Excluded patients with in-hospital mortality. Significant values are shown in bold. LOS, length of stay; OR, odds ratio; aOR, adjusted OR. Multivariable regression adjusted for variables that were significant in univariate regression model in Appendix A. ^b^ LOS > 9 days. *p*-values < 0.05 are shown in bold.

**Table 3 jcm-13-00772-t003:** Associations between study variables and postoperative complications.

Outcomes	Robot-Assisted vs. Pure Laparoscopic			
	Univariable		Multivariable	
	OR (95% CI)	*p* Value	aOR (95% CI)	*p* Value
Complications (any)	0.99 (0.78, 1.27)	0.955	0.69 (0.54, 0.88)	**0.003**
Bleeding	1.17 (0.88, 1.57)	0.280	0.73 (0.54, 0.99)	**0.045**
Pneumonia	0.48 (0.27, 0.84)	**0.011**	0.49 (0.28, 0.86)	**0.013**
Infection	0.81 (0.54, 1.22)	0.316	0.55 (0.36, 0.85)	**0.007**
Wound complications	0.47 (0.30, 0.72)	**<0.001**	0.33 (0.20, 0.54)	**<0.001**
Sepsis	0.48 (0.34, 0.67)	**<0.001**	0.49 (0.34, 0.69)	**<0.001**

Significant values are shown in bold. LOS, length of stay; CCI, Charlson Comorbidity Index; OR, odds ratio; aOR, adjusted OR. Multivariable regression adjusted for variables that were significant in univariate regression model in Appendix A.

## Data Availability

The datasets used and/or analyzed during the current study are available from the corresponding author on reasonable request.

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
