# Peer review of "Inpatient Outcomes of Patients Undergoing Robot-Assisted versus Laparoscopic Radical Cystectomy for Bladder Cancer: A National Inpatient Sample Database Study"

_jcm, 2024, doi:10.3390/jcm13030772_

Round 1
Reviewer 1 Report
Comments and Suggestions for Authors
The authors conducted a comparative analysis of outcomes between laparoscopic radical cystectomy (LRC) and robot-assisted radical cystectomy (RARC). The study utilized data from over 3000 patients extracted from the US National Inpatient Sample (NIS) database. The findings suggest that robot-assisted radical cystectomy is associated with a 35% reduced risk of unfavorable short-term outcomes, encompassing in-hospital mortality, prolonged length of stay (LOS), and postoperative complications when compared to pure laparoscopic radical cystectomy.
However, several points deserve consideration.
Given the widespread adoption of robotic platforms and the inherent technical challenges of the pure laparoscopic approach, the implications of this study may primarily serve as food for thought rather than directly influencing clinical practice, as the findings may not be timely.
Notably, the study lacks information about the type of urinary diversion employed. Equally important is the absence of details regarding the techniques used for diversion configuration—whether they were totally intracorporeal or extracorporeal. Moreover, there is a dearth of information about postoperative care, such as Enhanced Recovery After Surgery (ERAS) protocols.
The enrollment period appears to be broad, with the possibility that older cases were performed using the pure laparoscopic method, while more recent cases utilized a robotic approach. While acknowledging the challenges in retrieving such information, it is crucial for readers to have these details, as they significantly impact the interpretation of the study's results.
Author Response
While we acknowledge the concerns regarding the widespread adoption of robotic platforms and the potential technical challenges associated with pure laparoscopic approaches, we believe our study adds valuable insights into the comparative outcomes of these surgical techniques. The observed reduction in unfavorable short-term outcomes with robot-assisted radical cystectomy suggests that exploring the nuances of different approaches remains pertinent.
We agree with the reviewer about the importance of specific details on urinary diversion type and configuration techniques, as well as information about postoperative care. Regrettably, we were not able to include this information since they are not available in the dataset. We have added these issues as one of the study limitations to remind the readers.
We also agree that the enrollment period was broad, and surgical technique and quality of care may evolve of over time. More older cases were performed using the pure laparoscopic method, while more recent cases using robotic approaches, which may bias the results. Accordingly, we included ‘admission year’, categorized into three intervals (2008-2010, 2011-2014, and 2015-2018), as one of the covariates to be adjusted for in the multivariable regression analyses. Please check the revised tables and Results.
In addition, we've streamlined the tables in the main text for simplicity, while the comprehensive table has been relocated to the supplementary materials. We believe this would largely improve the readability and clarity of the paper.
Reviewer 2 Report
Comments and Suggestions for Authors
Bladder cancer (BCa) is one of the most common tumors of the urinary system, as well as the ninth most common cancer worldwide. Of bladder cancers, 75% are non-muscular invasive bladder cancer (NMIBCa). In the remaining 25–30% of patients, BCa has already invaded deeper layers of the bladder wall (MIBCa—muscle-invasive disease), that requires more radical treatment such as cystectomy. Advances in surgical equipment and techniques has led to increased use of minimally invasive radical cystectomy in the surgical treatment of bladder cancer.
The aim of this study is to compare the inpatient outcomes of patients with bladder cancer who underwent either robot-assisted or laparoscopic radical cystectomy, utilizing a nationally representative US inpatient database.
Using data from US National Inpatient (NIS) database, the authors found that in patients with bladder cancer undergoing minimally invasive cystectomy, robot-assisted radical cystectomy is associated with reduced risk for in-hospital mortality, prolonged LOS, and postoperative complications, including pneumonia, sepsis, and wound complications, compared to pure laparoscopic radical cystectomy.
This article is important because shows a clear advantage in post-surgical outcome of patients who underwent robot-assisted procedure, with a 50% lower risk in-hospital mortality. These findings highlight the advantages of robotic-assisted minimally invasive procedures over pure laparoscopic procedures performed for bladder cancer.
However, the study has some limitations. First, the database did not include the pre-operative status, crucial for the surgical outcomes. In addition, it would be interesting to collect data from these patients after discharge from the hospital, in a proper (at least one year long) follow up, to better understand the outcomes of the two different surgical procedures.
I suggest adding the following scientific article links to the bibliography section for a more accurate representation of the references and the general topic of this study:
- 10.1111/bju.16210
- 10.3390/cancers14102545
- 10.23736/S0393-2249.20.03850-3
Comments on the Quality of English Language
Minor editing
Author Response
Thank you for the insightful review and comments. We have added these references into citations. The limitation section has been updated to mention the importance to consider pre-operative status, given it is not available in the dataset. We share the view that collecting post-hospital discharge data would be valuable. Regrettably, the current NIS dataset does not encompass such information, as documented in the limitations section.
In addition, we've streamlined the tables in the main text for simplicity, while the comprehensive table has been relocated to the supplementary materials. We believe this would largely improve the readability and clarity of the paper.
Reviewer 3 Report
Comments and Suggestions for Authors
The authors aimed to compare in-hospital outcomes between robot-assisted and pure laparoscopic surgery performed for patients with bladder cancer. They concluded robot-assisted radical cystectomy is associated with reduced risk of unfavorable short-term outcomes, including in-hospital mortality, prolonged LOS, and postoperative complications compared to pure laparoscopic radical cystectomy in patients with bladder cancer.
It is an interesting topic.
In terms of historical background, open surgery, laparoscopic surgery, and robot-assisted surgery have appeared in that order. If we simply compare the results of the different techniques, the results of the more recent techniques should naturally be better.
Therefore, it should be assumed that the results of ‘robot-assisted surgery performed in recent years’ have improved compared to those of ‘laparoscopic surgery performed previously’.
If the results of this procedure were purely comparative, the analysis should include additional historical background factors.
Author Response
We share the reviewer's insight into the ongoing advancements in surgery technique, quality and post-surgical care over time, which might influence the study findings. Indeed, our study did not include such historical or temporal considerations. Accordingly, we added ‘admission year’, categorized into three intervals (2008-2010, 2011-2014, and 2015-2018), as one of the covariates to be adjusted for in the multivariable regression analyses.
Please check the revised tables and Results. Nevertheless, we have also highlighted this in the limitations section, aiming to remind readers to consider this issue when interpreting the results.
In addition, we've streamlined the tables in the main text for simplicity, while the comprehensive table has been relocated to the supplementary materials. We believe this would largely improve the readability and clarity of the paper.

Round 2
Reviewer 1 Report
Comments and Suggestions for Authors
Dear Authors,
I acknowledge and appreciate your efforts, but the raised concerns have not been addressed.
Reviewer 2 Report
Comments and Suggestions for Authors
Authors answered all comments and suggestions.
Comments on the Quality of English LanguageMinor editing.
Reviewer 3 Report
Comments and Suggestions for Authors
Previously identified problems have been corrected appropriately.